# Modeling quarantine during epidemics and mass-testing using drones

**Leonid Sedov**[1]☺*, **Alexander Krasnochub**[2]☺, **Valentin Polishchuk**[1]☺

**1** Communications and Transport Systems, ITN, Linköping University, Norrköping, Sweden, **2** Joint Institute for High Temperatures, Moscow, Russia

☺ These authors contributed equally to this work.
* leonid.sedov@liu.se

**Data Availability Statement:** All relevant data are within the manuscript.

**Funding:** The author(s) received no specific funding for this work.

## Abstract

We extend the classical SIR epidemic spread model by introducing the "quarantined" compartment. We solve (numerically) the differential equations that govern the extended model and quantify how quarantining "flattens the curve" for the proportion of infected population over time. Furthermore, we explore the potential of using drones to deliver tests, enabling mass-testing for the infection; we give a method to estimate the drone fleet needed to deliver the tests in a metropolitan area. Application of our models to COVID-19 spread in Sweden shows how the proposed methods could substantially decrease the peak number of infected people, almost without increasing the duration of the epidemic.

## Introduction

Importance of proactive COVID-19 screening, including symptomless people, has been acknowledged [1–6]. However, mass-testing may be seriously impeded by population's fear of visiting testing facilities due to potentially high concentration of infection there; the fear is confirmed by health officials who advise against visiting hospitals "unless necessary" (examples of such directives from authorities during COVID-19 pandemic abound): "Stay home" is the overarching recommendation during pandemic virtually everywhere in the world. The good news is that COVID-19 test does not have to be necessarily conducted at a designated facility because (despite being somewhat unpleasant,) the test can be self-administered: a person may collect the material him/herself or with the help of a family member. Still, having people go somewhere to pick up and drop the tests would beat the purpose of the social distancing. A possible solution is to use drones to distribute tests to the population as well as to collect the tests back, bringing them to laboratories; the test results could then be communicated back to people electronically, so that those with positive tests put themselves into quarantine. Here we follow the infection dynamics by extending the SIR model [7] to include the compartment for quarantined population and show how the testing intensity, increased with use of drones, decreases the epidemic spread. Since pre-symptomatic infectiousness makes early surveillance and control crucial (symptom-based actions are not as effective), our results may guide the cooperation between health and transportation authorities towards optimal use of the available resources (other transportation-related work in public health domain includes [8–10]).

**Competing interests:** The authors have declared that no competing interests exist.

## Methods

We consider a laboratory for COVID-19 tests processing, assigned to mass-test the population living in a certain service area: for a city with a single hospital (as e.g., in Norrköping—our running example) the area is the whole city, while more generally the region may be split among several laboratories (including, possibly temporarily set up) by the authorities (in the centralized model, considered here, it is the authorities who decide for each laboratory from where to collect the tests). To define the locations for test delivery/collection, use the population data with fine granularity, providing population count in every grid square (we used the map giving the number of people living in each 100m x 100m square [11]). The drone fleet is stationed at the hospital (the fleet may contain the hospital's own vehicle plus possibly machines subcontracted from drone service suppliers—we do not go into the economic details here, envisioning that the costs saved for the economy thanks to the slower virus expansion will outweigh the costs for renting drones services). The number $N$ of drones in the fleet and the capacity $C$ (the number of tests that can be loaded onto one drone) are our variables (representing the resources)—we present the number of days $D$, needed to collect tests from the whole city population, as the function of $N$ and $C$ (for visualization, in Fig 1 we present the a separate $N$–$D$ curve for each considered value of $C$).

In the output we seek a route for each drone (see Fig 2 for an example), with the indication of how many people the drone serves in each square along its route (in some squares this number may be 0 meaning that the drone just passes over the square); the total number of people served by the route between successive returns to the hospital should not exceed the drone capacity $C$. Note that a square may be served by several drones; e.g., if the number of people living in a square is $2C$ or more, the square will have to be visited at least two times (or possibly more, if one of the times the drone is not used in the square to its full capacity). We suggest that in practice the drone repeats the route twice: first to distribute the tests and then to collect them—the interval between the distribution and collection does not influence our results since we compute a single route for every drone and duplicate it, thus simply doubling the length of the routes and consequently the number of drones needed to maintain certain testing frequency (if the interval is long, people may store their specimen in the fridge [12]). The time a

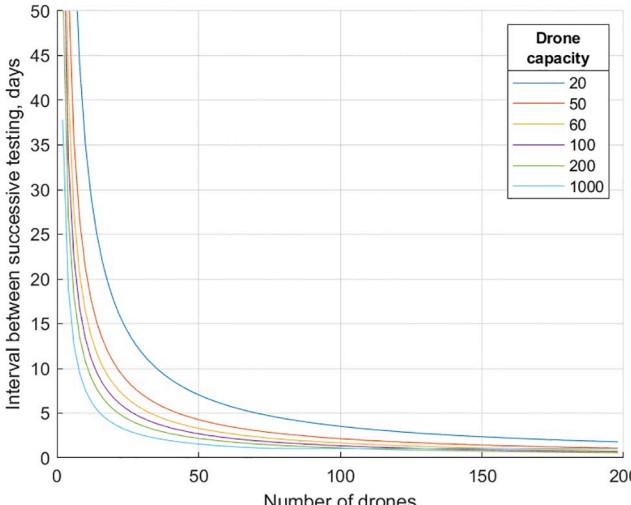

**Fig 1. The number of days $D$ needed to collect tests from the whole city population as the function of the number $N$ of used drones (of different capacities).**

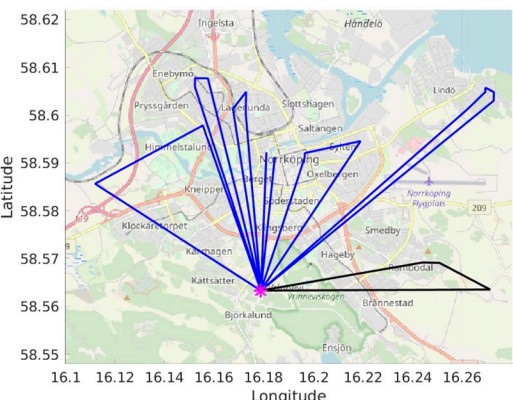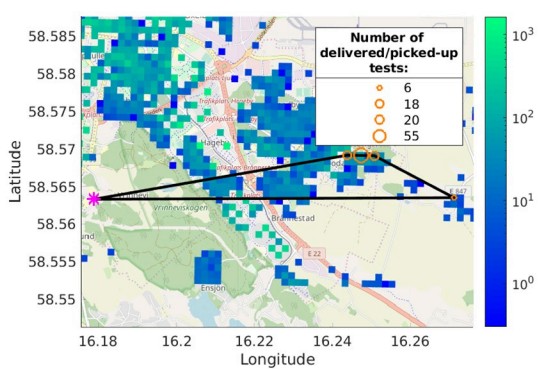

**Fig 2. Drone routes.** Left: An example route of a drone with capacity 100; one of the tours in the route is shown black. Pink asterisk is the hospital. Right: A zoom in on the black tour: the orange circles depict the tests delivery/pickup locations; circle size is proportional to the number of tests delivered/picked-up at the location (99 tests are delivered on this tour). The underlying heatmap is the population density. The background map was rendered on the authors' machines from OpenStreetMap data [13] with OpenStreetMap Carto style [14] using the code available from [15].

drone spends in each square it serves (i.e., in each square where its route serves >0 people) is set to 15 min: this includes the time to possibly distribute the tests to (or collect the tests from) several cottages in the square; it also includes the time to change drone batteries at the base (hospital, lab) when needed.

The route for each drone consists of several tours, where each tour is a closed loop that starts and ends at the hospital and the number of people served on each tour is at most $C$ (any such tour may be executed by a drone without returning to the base as long as the battery capacity is not a constraining factor: for a drone with our maximum considered capacity $C = 1000$ this would hold in a reasonable city scenario because 1000 people may be typically reached on a tour of length 10-20km, which is below the range of a non-toy drone; in a rural scenario, however, where visiting 1000 people may require longer tours, the battery may become the limiting factor and the maximum tour length constraint will have to be added when computing the tours). Minimizing the total length of such tours, needed to cover all people in the area, is known as the Capacitated Vehicle Routing Problem (C-VRP). We used Google's OR-Tools [16] to solve the C-VRP, i.e., to find the tours that collectively serve all people in the area (for an indication of the solution quality, we remark that the OR-Tools output 1042 tours for capacity-100 drones, which is close to the minimum $102638/100 \approx 1027$ tours needed to serve the total of 102638 people living in our city with capacity-100 drones). Computing the tours for one value of the drone capacity $C$ took $\sim 4800$ sec on a computer with AMD CPU with 6 cores (12 threads) and base clock 3.4GHz.

Finally, to minimize the length $L$ of the longest drone route (which directly impacts the makespan, i.e., the time to fly all routes and thus serve the entire population), we split the tours into $N$ sets (we did this for every value of $N$ in a range), so that every set becomes a route for a drone (the tours in the drone's set are executed one-by-one in an arbitrary order). When splitting the tours, the objective is to minimize the length of the longest route (i.e., to minimize the maximum total length of the tours in a route)—it is the longest route that is the "bottleneck" of our solution, defining the longest time that a person has to wait for the drone. This is exactly the Minimum Makespan Scheduling problem aka Load Balancing (because it balances the "load" of every drone) aka Multiprocessor Scheduling (the drones can be viewed as processors which have to collectively process the set of tasks—fly all given tours). We formulated the

problem as an Integer Program (IP) and solved it using Gurobi solver [17] (separately for each value of $N$).

Specifically, in the IP the binary decision variable $x_{dt}$ was equal to 1 if the route for drone $d$ included the tour $t$. The IP minimized the length $L$ of the longest route (i.e., the objective function was $L$) subject to the constraints $\sum_d x_{dt} = 1$ for every tour $t$ (meaning that every tour must be included in exactly one route) and the constraints $\sum_l l_t x_{dt} \leq L$ for every drone where $l_t$ is the length of the tour $t$ (meaning that $L$ is the length of the longest route, i.e., the length of any route does not exceed $L$). The full IP model is thus as follows:

$$min \ L$$

$$subject \ to :$$

$$\sum_d x_{dt} = 1 \ \forall t$$

$$\sum_l l_t x_{dt} \leq L \ \forall d$$

We solved the above IP using Gurobi optimization software installed on Tetralith cluster [18] of Intel HNS2600BPB nodes with 32 CPU cores, provided by the Swedish National Infrastructure for Computing (SNIC). Computations took from $\sim 0.01$ sec (for $N = 2$, $C = 1000$) to $\sim 460$ sec (for $N = 54$, $C = 1000$) with the absolute IP optimality gap parameter set to 500 (i.e., the solver terminates when the solution objective is guaranteed to be within 500 seconds from the optimal).

To convert the longest route length $L$ into the time needed to fly all the routes (the makespan), we assumed that the drones operate 12 hours per day and fly with the speed of 60km/h. We do not investigate other values of these two parameters because the time of serving the routes scales linearly with them: e.g., if the drones fly 24/7 (the authority shall decide whether it is wise to disturb people's sleep and distribute the tests also overnight in order to increase the testing intensity) or with twice the speed, the service time simply halves, etc. We remark that, on the contrary, the dependence on the drone capacity $C$ is not straightforward at all: changing $C$ changes already the tours (in a way which is hard to predict a priori—the tours are output by the advanced Google OR-Tools optimization software), which of course influences the routes and their lengths. We therefore run our experiments separately for several values of $C$ (for each $C$, we solve the Minimum Makespan Scheduling problem for each value of $N$). Refer to Fig 1 for the results showing the relation between $N$ and the service time $D$.

Our code is available at https://github.com/undefiened/corona_drones for any community to make their own estimates like ours. The needed data is the laboratories locations (for the case of multiple laboratories, also the service area of each laboratory), the population density map and the drone capacity. Our GeoGebra applets for the SIQR model (described below) are accessible online at http://tiny.cc/SIQR (the generic SIQR model) and http://tiny.cc/SIQR_Swe (with Sweden's data plotted).

## Results

Recent advances in the drone (unmanned aerial vehicles, or UAV) technology allow one to perform unmanned delivery-to-the-door of various goods [19]. Drones succeed with blood transportation [20–24] (and even organ transportation is being explored [25]), implying that transporting by drones the less sensitive [12] COVID-19 test samples (e.g, upper respiratory specimens in nasopharyngeal and oropharyngeal swab or lower respiratory specimens like

sputum) may also be technically feasible [26] (related studies went so far as to explore testing for avian influenza A (H7N9) virus directly on the drone [27]). This opens the potential to use drones for mass-testing the population for COVID-19, most importantly—including asymptomatic people: it has been reported that symptomless people can be infectious as well [1–3, 5] (in fact, since symptomatic people were advised to isolate very early, it may be the case that the pandemic occurred mostly due to asymptomatic transmission [4]). In particular, if the infected but asymptomatic people become infectious after $D = 4$ days from being infected [2], then (assuming the tests can determine that a person is infectious starting from day 0 of the infection) quarantining everyone within 4 days of the infection could stop asymptomatic COVID-19 spread altogether (assuming responsible behavior of self-quarantining from the asymptomatic people who received positive test results). We estimate that for a medium-size city with ∼100000 inhabitants (Norrköping, Sweden—our guinea pig), 36 Switzerland Matternet [28] drones (each carrying 100 tests) suffice to visit everyone (distributing, collecting and returning the tests to the lab) once every 4 days—see Fig 1 which shows our results for drones of various capacity and for varying number of drones. (See Methods section for details of obtaining the presented results.)

To put our estimates into practical perspective, we connect to existing drone models. The weight of a single test (20g) is calculated as the sum of the viral transport medium weight (3g [29]), the plastic tube with the cup and the swab (7g = weight of a 10mL syringe [30]), an A4 list of instructions (5g) and packaging (5g). Since a single test would fit into ∼11cm x 2cm x 2cm box, 100 tests would occupy 4-5L of cargo. This is at the upper limit of the volume for some of the drone models (e.g., the one from Amazon Prime [31]), implying that cargo volume may be a capacity-limiting factor. As far as the payload weight goes, the market essentially offers either vehicles with few kilograms of payload (such as Amazon Prime drones [31], or medical delivery drones from UPS [32] or Switzerland Matternet [28]) or large machines lifting hundreds of kilograms (such as Boeing Cargo air vehicle [33] or ACC air drone [34]). Thus in terms of weight, different drones may carry anywhere between ∼50 to ∼5000 tests; for conservative estimates, we experimented with the capacity ranging from 20 to 1000. (The different drone models also have slightly different maximum speeds, but they differ by at most a factor of 2, and 60km/h is a reasonable speed for all the drones—our estimates scale linearly with the speed, so it is straightforward to adjust our results for other flying speeds.)

## SIQR: An extension of the SIR model

For the impact of our results on epidemic spread theory, we introduce an extension of the SIR model for epidemic development [7, 35]. The vanilla SIR model's transitions between its 3 compartments (or, states) $S$, $I$, $R$ (Susceptible, Infected, Recovered resp.) are governed by the following differential equations

$$dS/dt = -aSI$$

$$dI/dt = aSI - bI$$

$$dR/dt = bI$$

where $S$, $I$, $R$ stand for the number of susceptible, infected, recovered people resp. as functions of the time $t$ (the notation is slightly abused by identifying the numbers with the compartment names), and $a$, $b$ are the parameters signifying the rates of the transfer between the states. The parameters are estimated from the clinical data; e.g., $b = 1/T$ where $T$ is the average duration of the disease in a patient.

For a theoretical application of our estimates, we restate our vision that each drone will fly its route continuously: that is, even though we are solving a "single-shot" service problem which minimizes the time needed to serve every inhabitant only once, we envision that people will be tested periodically, with the interval $D$ between the tests (the recurring testing is needed to catch those who were infected after the previous test). We introduce the "Quarantined" compartment ($Q$) into the SIR model and assume that $I$ now represents infected but not quarantined people (i.e., those spreading the disease); we dub the extended model SIQR. (We remark that our model is different from other extensions of SIR such as SEIR [36–38] introducing Exposed people or models with vaccination [39] which either decrease $S$ with pro-active vaccination or move people directly from $S$ to $R$ using the reactive vaccination during the epidemics.) Transitions to $Q$ happen only from $I$, and the rate of the transition is $1/D$ (this is where our estimates of $D$ are used); transitions from $Q$ happen only to $R$ with the rate $b$ (the same as from $I$ to $R$—we assume that the removal of an infected patient does not depend on whether the patient is quarantined or not). Thus the state transitions in SIQR model are governed by

$$
\begin{aligned}
dS/dt &= -aSI \\
dI/dt &= aSI - (b+c)I \\
dQ/dt &= cI - bQ \\
dR/dt &= b(I+Q)
\end{aligned}
$$

where $Q$ is the number of people on quarantine and $c = 1/D$.

Since the parameters of SIR (and hence SIQR) vary widely depending on the country, anti-epidemic measures, quality of the available data and other factors, we do not confine ourselves to specific values for the parameters. Instead, we present SIQR curves for a whole range of the parameters. Specifically, we started from the open-source GeoGebra applet [40] which computes SIR curves $S$, $I$, $R$ for any values of the parameters $a$ and $b$ set by the user; the parameters are set simply by moving $a$ and $b$ sliders, so the curves change interactively as the parameters are modified (the applet assumes that the total population size is 1, i.e., it shows the fractions of population in the $S$, $I$ and $R$ compartments). We changed the sliders to represent the recovery time and the effective reproduction number $R_0$ (the number of people infected by one infected person): $a$ and $b$ are then calculated as $b = 1/T$, $a = R_0$, $b = R_0/T$; we choose $T$ and $R_0$ as the user-input parameters because they are the ones estimates for which are easier to find reported (sources with various estimates bound, but we do not restrict ourselves to any particular one since we give the full flexibility with the parameters choice). To extend SIR to our SIQR, we add the slider for the inter-testing interval $D$, calculate $c = 1/D$ and change the SIR differential equations to the SIQR equations above. We also depict the curve $CI$ showing the total (cumulative) number of people infected; that is, $CI(t)$ is the number of people that have been infected by the time $t$ (since the transition to infected happens only from the susceptible, $CI$ is simply the total population minus $S$). Fig 3 shows screenshots of our SIQR applet—it can be seen how the curve flattens with $D = 8$ (the applet is interactive, and we invite the reader to play online with our sliders for the different parameters at http://tiny.cc/SIQR).

Fig 4 shows real data (pink crosses) for confirmed COVID-19 cases in Sweden starting from 100 cases on March 06 2020 [42] (see [43, 44] for detailed COVID-19 history and projection for ICU beds demand in Sweden resp.). The basic SIR model ($Q = 0$) fits the data with a modest (for the novel coronavirus) value of $R_0 = 2.27$, which is reasonable given the general hygiene and cultural distancing in Sweden. Because not all the population was tested, we do

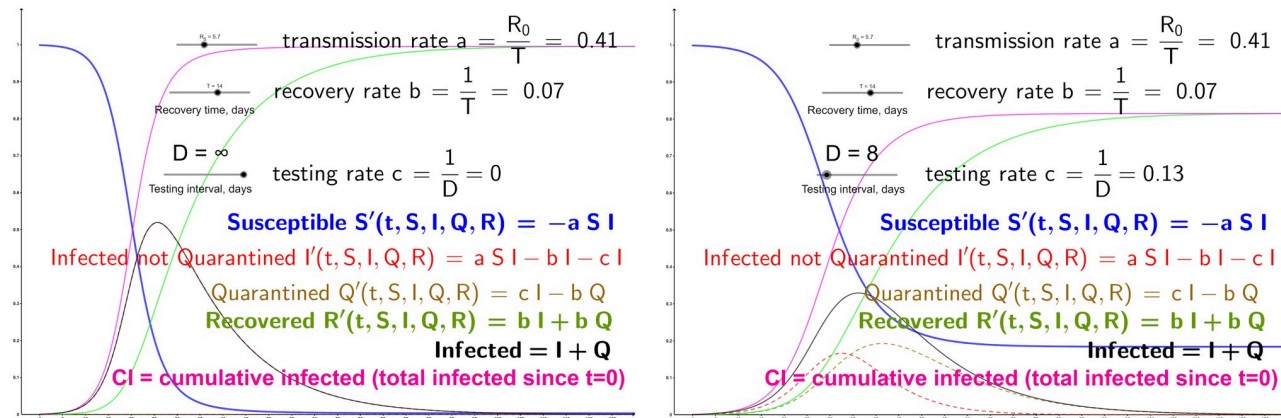

**Fig 3. Snapshots of our online applet showing the curves for quite high $R_0$ = 5.7 [41].** Left: SIR curves ($Q$ = 0, obtained by setting $D$ to infinity). Right: SIQR with the testing interval $D$ = 8 (see Fig 1 for the number of different-capacity drones needed to reach $D$ = 8).

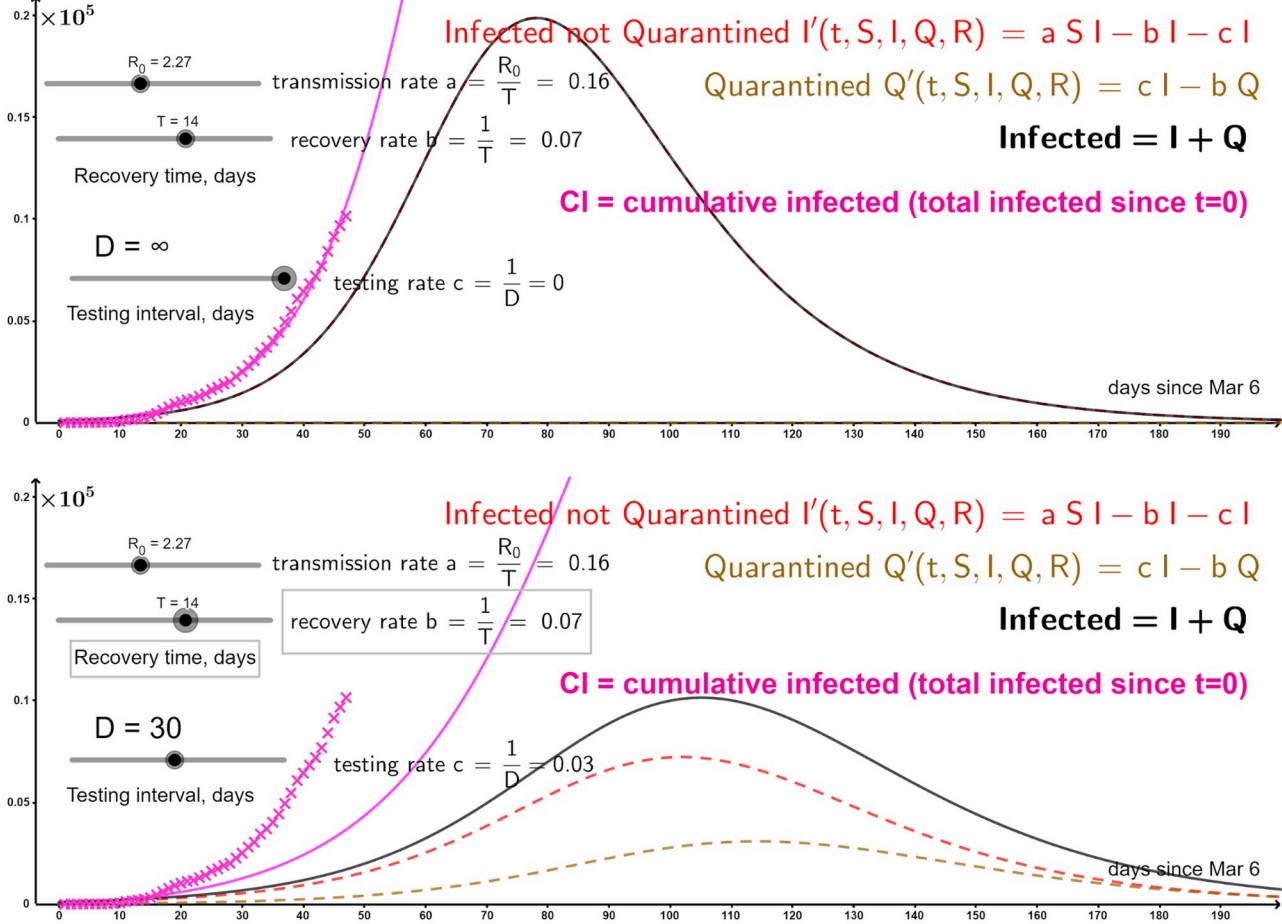

**Fig 4. COVID-19 cases growth (starting from 100 cases) in Sweden [42] shown with pink crosses.** Top: SIR model ($S$ and $R$ not shown). Bottom: Mass-testing even as rarely as every $D$ = 30 days flattens the curve.

not know the true number of infected people. Instead of trying to estimate this true number (which would introduce yet another parameter), we fit our pink curve into the confirmed cases; in this sense $R_0$ represents here the number of new confirmed cases per one confirmed case. Our SIQR model suggests that regular mass-testing with the interval $D = 30$ (which roughly amounts to randomly testing $\sim 3.3\%$ of the population every day) would flatten the curve quite significantly. We again invite the reader to play with our interactive GeoGebra applet http://tiny.cc/SIQR_Swe to see the effects of the testing frequency, as well as the changes in the parameters $R_0$ and $T$.

## Discussion

Our work may help the authorities to quantify the lessons learned during the COVID-19 pandemic and utilize them during future epidemics (more generally, our methods may be applied to any kind of mass delivery and collection: e.g., the drones may distribute immunity tests to release people from quarantine). Indeed, one stumbling block to drones ubiquity is the generally absent regulation (in particular, in regions with less strict or absent (anti-)drone laws, UAVs are already widely used for medical applications including infectious disease surveillance and epidemiology [45–48]). With a proactive thinking, the authorities could design sets of regulations with different levels of strictness: in situations like epidemics, more lenient regulations could take force and let the drone operations rise to higher levels than during nominal course of events—the switch may be justified not only by the extreme social value of the drones use (as shown here), but also by the fact that during the (even partial) quarantine there are fewer people on the streets, which lowers the ground risk of drone operations (one of the concerns for the regulators [49–54]).

We emphasize that our model of mass-testing can be applied at any geographic scale. In particular, while the epidemic is confined to a limited area, it may be reasonable to concentrate on mass-testing only the potentially affected region. As far as our theoretical results (SIQR) are concerned, incorporating people on quarantine into epidemic spread modeling may not only provide more accurate prediction of the disease spread, but also help understand the economic consequences of the quarantine (see [55] for discussion of COVID-19 economic impact).

## Conclusions

We introduced an extension of the SIR model for epidemic spread. Our SIQR model adds to SIR the compartment Q representing quarantined population. Sending people into quarantine is a well known anti-epidemic measure, and SIQR model allows one to quantify its effects. We also studied the use of drones for regular mass-testing of the population, and presented algorithms for estimating how many vehicles are needed to deliver tests in a metropolitan area.

Future research should explore economic considerations underlying the use of drones for mass testing. Answering questions like how much could such a service cost and who will pay for it are crucial for a potential real-world implementation. Among the supporting services, the most important one is the provision of energy supply for the drones via charging stations or battery replacement docks.

One technical limitation of our research is the assumption that all vehicles have the same capacity. In reality, it is more likely that many different drone models will be used for the mass testing. In addition to the UAVs possibly owned by the hospital, the drones may be leased from local businesses, introducing even a higher variety into the fleet. The algorithms in this paper work with uniform drone capacity only, and extending them to varying capacities is an open problem.

Another open research direction is assigning the population to the test facilities. Our methods work for a single facility, and it would be interesting to address the question of optimal splitting of the area among the test centers. Last but not least, many other facility location problems open up: where would it best to place a new test processing laboratory, how many labs are needed for a given area, etc.

## Author Contributions

**Conceptualization:** Leonid Sedov, Alexander Krasnochub, Valentin Polishchuk.

**Data curation:** Leonid Sedov, Alexander Krasnochub, Valentin Polishchuk.

**Formal analysis:** Leonid Sedov, Alexander Krasnochub, Valentin Polishchuk.

**Investigation:** Leonid Sedov, Alexander Krasnochub, Valentin Polishchuk.

**Methodology:** Leonid Sedov, Alexander Krasnochub, Valentin Polishchuk.

**Software:** Leonid Sedov, Alexander Krasnochub, Valentin Polishchuk.

**Visualization:** Leonid Sedov, Alexander Krasnochub, Valentin Polishchuk.

**Writing – original draft:** Leonid Sedov, Alexander Krasnochub, Valentin Polishchuk.

**Writing – review & editing:** Leonid Sedov, Alexander Krasnochub, Valentin Polishchuk.

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
